# Social Processes of Young Adults’ Recovery and Identity Formation during Life-Disruptive Mental Distress—A Meta-Ethnography

**DOI:** 10.3390/ijerph20176653

**Published:** 2023-08-25

**Authors:** Ida Marie Skou Storm, Anne Kathrine Kousgaard Mikkelsen, Mari Holen, Lisbeth Hybholt, Stephen Fitzgerald Austin, Lene Lauge Berring

**Affiliations:** 1Research Unit, Mental Health Services East, Psychiatry Region Zealand, 16, 4000 Roskilde, Denmark; annekath@gmail.com (A.K.K.M.); lihy@regionsjaelland.dk (L.H.); sfa@regionsjaelland.dk (S.F.A.); 2Institute of Regional Health Research, University of Southern Denmark, 55, 5230 Odense, Denmark; lelb@regionsjaelland.dk; 3Psychiatric Research Unit, Psychiatry Region Zealand, 6, 4200 Slagelse, Denmark; 4Health and Society, Department of People and Technology, Roskilde University, 1, 4000 Roskilde, Denmark; holen@ruc.dk; 5Institute for Psychology, University of Southern Denmark, 55, 5230 Odense, Denmark

**Keywords:** recovery, mental health, young adult, belonging, friendship, systematic review, meta-synthesis

## Abstract

Young people’s mental health recovery is well-explored in empirical research, yet there is a lack of meta-studies synthesizing the characteristics of young people’s recovery. This meta-ethnography explores young adults’ recovery during life-disruptive experiences of early psychosis or schizophrenia. Based on a systematic literature review search, 11 empirical qualitative studies were included for synthesis. Inspired by young people’s prominent experience of social isolation in the included studies, we applied an interpretive lens of belonging deriving from the sociology of youth. The synthesis presents five themes: (1) expectations of progression in youth in contrast with stagnation during psychosis, (2) feeling isolated, lost and left behind, (3) young adults’ recovery involves belonging with other young people, (4) forming identity positions of growth and disability during psychosis, and the summarizing line of argument, (5) navigating relational complexities in the process of recovery. While suffering from social isolation, young people’s recovery is conceived as getting on with life, like any other young person involving connecting and synchronizing life rhythms with their age peers. Socializing primarily with caring adults entails being stuck in the position of a child, while connecting with young people enables the identity positions of young people. This synthesis can inspire support for young people’s recovery through social inclusion in youth environments.

## 1. Introduction

The personal and social processes of mental health recovery have been the subject of research for decades [1,2,3,4,5]. Although the research indicates that the vast majority of long-lasting mental distress is already present in young adulthood [6,7,8], only in recent years have researchers focused on exploring particular characteristics of young people’s recovery [9]. The current study contributes to the qualitative knowledge in this field by generating a deeper insight into young people’s recovery in a period of life when they explore and pursue who and what to become. Taking inspiration from Arnett’s concept of emerging adulthood [10], this study adopts a relatively broad understanding of being young. Arnett’s concept of emerging adulthood implies an extension of the formative period of early life into young people’s mid-twenties. This extension is due to cultural changes in the Western world involving a longer education and postponement of independent living and family building. Furthermore, experiences of life-disrupting mental distress tend to delay educational attainment and family-building [8], leading to a further prolongation of this groups’ transition to adult living. Hence, in this study, we explore the recovery experiences of young adults.

Several recent studies of young people’s recovery argue that elements from the adult literature, such as connectedness, hope and optimism about the future, identity, meaning and empowerment [2], are also present in young people’s recovery [9,11,12]. The studies on young people’s recovery argue, however, that this group expresses more dynamic and fluctuating perceptions of recovery [9,11], possibly due to young people’s polarized needs of independence and support [11]. Furthermore, studies agree that young people’s recovery journey, which takes place as they are figuring out who to become, is a highly interactive process, highlighting the great importance of social interaction, participation and agency in young people’s recovery [9,11,12]. Therefore, in this synthesis, we understand young people’s recovery as being an individually experienced yet highly socially negotiated processes, in accordance with perspectives that emphasize the social nature of recovery [3,4,5,13].

Two recent studies show a discrepancy between how young people understand their own situation in times of life-disruptive mental distress and how they are understood by mental health services, in example when assessed via psychiatric or psychological scales [14,15]. While Topor laments that patients’ power of reasoning has generally been considered impaired in psychiatric contexts, he conveys that patients possess a wealth of knowledge about their situation and recovery, which is of great worth to the development of mental health services [4]. In alignment, Slade states that recovery is best judged by the person living with the experience [16] and Tew argues that listening to the stories of people in mental distress develops awareness and decreases the tendency to see them as ‘others’ [5]. This highlights the importance of exploring young people’s lived experiences of mental distress and recovery. Many empirical studies with this focus have been published in recent decades. This enables qualitative meta-studies to be conducted, gathering interpretations to produce a nuanced meta-interpretive insight into young people’s experiences with mental health distress and recovery. As we have found very few qualitative meta-studies in this field [9], this meta-ethnography contributes by synthesizing studies of young people’s recovery experiences during times of highly life-disruptive mental distress. To explore this, we selected studies of young people diagnosed with psychosis. In a psychiatric context, psychosis is regarded as an early possible sign of schizophrenia, which is a highly life-disruptive condition involving changed perceptions of oneself and the world, potential educational, vocational and social detachment, experiences of stigmatization, and, at worst, increased risk of premature death [17,18] due to somatic diseases and suicide [19]. A recent review emphasizes that young people experiencing psychosis are particularly preoccupied with developing a positive view of self on their path to recovery [20]. Conversely, an empirical study of older people’s recovery suggests that this group is less engaged in identity work at their later stage of life [21]. This emphasis of young people’s focus on identity is in alignment with studies describing how experiences of psychosis and schizophrenia have been associated with a lost sense of self in a social world [22]. Hence, the meaning of young people’s identity formation appears relevant to a synthesis of young people’s recovery during experiences of psychosis or schizophrenia.

Inspired by the salience of being socially isolated and left behind, which we inductively generated through a literature synthesis, we employ a theoretical lens of young people’s ‘belonging work’ deriving from the sociology of youth [23]. This is consistent with studies of young people’s recovery arguing that recovery and identity formation are highly interactional processes in youth and young adulthood [9,11,12].

The aim of this meta-etnography is therefore to synthesize interpretations of young adults’ recovery and identity, explored through the lens of young people’s belonging work in order to answer the research question: What characterizes young people’s experiences of recovery and identity captured through a synthesis of qualitative empirical studies of young people diagnosed with psychosis or schizophrenia?

## 2. Methods

This study positions itself within the interpretivist paradigm; hence, we chose an interpretive type of meta-analysis. In contrast to narrative or systematic reviews, which are aggregative in nature, the purpose of interpretive literature syntheses is to develop meta-analytical qualitative explanations that are innovative in nature. The meta-ethnographic synthesis is a well-developed method involving inductive comparative processes of translation. This is appropriate when qualitative empirical studies have thoroughly explored a specific field of interest [24]. The approach appears relevant, as we found no other interpretive literature synthesis of young people’s recovery in psychosis. The principles of Noblit and Hare guided this meta-ethnography [25] in its aim to interpret a common storyline [26] across the included studies, and enable the interpretations from each study to be synthesized into a new, overall social explanation. The method involves seven phases, which are as follows: (1) getting started; (2) deciding what is relevant to the initial interest; (3) reading the studies; (4) determining how the studies are related; 5) translating the studies into one another; (6) synthesizing translations; and (7) expressing the synthesis [25]. The original method suggested performing ‘reciprocal’ translations of analogous findings and ‘refutational’ translations of conflicting findings, while suggesting the ‘lines-of-argument’ interpretation as an additional step to integrate all findings like pieces in a jigsaw puzzle. This meta-ethnography is inspired by France’s later integration of the lines-of-argument synthesis as a fixed component of the meta-ethnographic method [27]. To strengthen methodological transparency, we report on each phase, taking inspiration from France’s eMERGe reporting guidance [27].

### 2.1. Selecting and Locating Relevant Studies

Prior to the synthesis, a systematic literature search was conducted in March 2021. Assisted by an information specialist [28], the first author conducted a search involving a building block strategy [29] in seven databases between January and March 2021 (Figure 1). The search combined five search blocks. Four blocks, ‘youth’, ‘psychosis’, ‘recovery’ and identity’, were defined with reference to the aim and research question of the study, while the fifth block limited the search to qualitative studies. We identified search terms in the thesauruses of each database and free-text words through an examination of keywords of relevant publications. Free-text words were applied across databases. Given the study’s specific interest in young people, we exclusively searched for youth and free-text synonyms of this term in the title field. As young people’s experiences of recovery are an eligible subject of research within a range of disciplinary fields, the search was conducted in seven databases, including Medline, CINAHL, PsychInfo, Embase, SocIndex, Scopus, and Web of Science. Due to the broad perceptions of recovery and identity as both personal and social processes, various synonyms were used (Appendix A illustrates the search string from Embase in the Appendix A). The systematic literature search identified 3027 publications. Duplicates were removed, leaving 2093 references to be title-screened by the first and second author. The abstracts of the remaining 119 studies were evaluated against inclusion and exclusion criteria.

### 2.2. Criteria of Inclusion and Exclusion

Studies were included if they presented qualitative interpretations of recovery experienced by young people and were published in English or Nordic languages between the years 2000 and 2021. Studies were excluded if they were not qualitative and empirical, if they explored other perspectives than those of young people, or if the analyses combined accounts from young people and parents or mental health staff (Table 1). Subsequently, the first and second author assessed 22 full-text publications. Inspired by Campbell, we assessed (1) the relevance of the research to the synthesis topic [30], and (2) whether the publication reported interpretations from qualitative research, involved qualitative methods of data collection and analysis and presented a certain level of explanatory insight [31]. Publications were excluded if they did not contain interpretive findings that were relevant to the synthesis topic in terms of youth, psychosis, recovery and identity, or if they did not qualify as qualitative research. The assessment of the level of explanatory insight was inspired by Sandelowski and Barroso’s typology [31], enabling an evaluation of the degree of data transformation in qualitative studies. As ‘no finding publications’ and ‘topical surveys’ do not present qualitative interpretations, the assessment of explanatory insight led to further exclusion of two publications. The last author was consulted in case of doubt regarding the assessment of explanatory insight. 

Subsequently, the first author conducted a citation pearl-growing strategy, pursuing publications related to relevant posters from the systematic search as well as references of the included publications and relevant reviews. Four additional publications were included following an assessment of relevance to the synthesis topic and level of explanatory insight, resulting in the inclusion of 11 studies for synthesis. While meta-ethnographies can involve from 40 to 100 studies [32], we chose to include the 11 studies that honored the criteria defined above. Moreover, the wealth of information and the context of each publication may be lost with a larger number of studies [33], and the inclusion of 11 studies is consistent with the recommendation of using a maximum of from 10 [34] to 12 studies for synthesis [35].

### 2.3. Quality Appraisal

In order to refine the assessment, the quality of the 11 included studies was appraised by three authors, using 20 criteria developed by the Medical Sociology Group Committee [36]. The meta-ethnographic approach allows for a synthesis of qualitative studies with different research methodologies [33]. Hence, we chose a detailed appraisal instrument involving all 11 key criteria that Hannes et al. refer to in their comparative study of quality appraisal instruments [37]. Due to divergent opinions regarding what constitutes high-quality qualitative research [31,38], and to obtain insightful interpretations despite minor methodological flaws, publications were not excluded following quality appraisal [30,39]. Rather, the approach sharpened our attention to the methodological and analytical strengths and weaknesses of the included publications.

### 2.4. Reading the Studies

Guided by the chronological order of publication [39], the first, second and last author extracted key metaphors regarding young people’s recovery and identity processes from the included studies. To synthesize interpretations, we exclusively extracted second-order constructs, which comprise interpretations articulated by the authors of the included studies [26]; however, sometimes these articulations included participant quotations verbatim. Noblit and Hare’s metaphorical perspective on language enabled us to interpret how the extracted metaphors corresponded across the included studies, even when expressed in different words. Furthermore, this allowed us to rename and translate key metaphors during the process of synthesizing extracts into third-order interpretations [26]. The first, second and last author extracted metaphors from all sections of the included studies in order to achieve a comprehensive interpretation of each study. Extraction from the discussion sections was conducted with caution due to the subjective and speculative nature of these sections [33]. We applied two strategies of metaphor extraction. Firstly, taking inspiration from Sattar et al. [32], the first and second author inductively extracted metaphors from the headings and body of the studies. As a second strategy, the first and last author searched for interpretations of ‘youth’, ‘psychosis’, ‘recovery’ and ‘identity’ in each of the included studies in order to explore similarities and differences between interpretations. The synthesized studies are presented in Table 2.

### 2.5. Determining How Studies Are Related

The included studies were published between 2005 and 2021. All studies were conducted in a Western context, apart from a Chinese study providing insight into young people’s recovery in an Asian context [44]. The main discipline among the authors was psychology (*n* = 8), supplemented by studies within the fields of nursing, occupational therapy and psychiatry. The age-span of participants across studies was 16–34 years, with the vast majority being males aged 18–30 years. All participants underwent psychiatric treatment following first episode of psychosis or schizophrenia and were recruited for participation through mental health services. Participants shared their experiences in semi-structured individual or focus-group interviews from six months to five years following treatment onset, providing some participants with more time to gain recovery experiences than others. The majority of the studies contain little description of participants’ socioeconomic and cultural backgrounds. However, most studies either define participants’ experiences of being in recovery as an inclusion criterion [43], or exclude participants assessed to be severely affected by psychosis [40,46], or those whose condition was related to the use of drugs or alcohol [42,44,48]. In general, the population in the included studies appears less mentally and socially challenged than the excluded population. The foci of the included studies varied between capturing the experiences of specific groups, such as young men, teenagers and group-program participants, and capturing the experiences of hospital admissions as well as relationships with friends, close family and psychiatric staff. It was possible to construct a common storyline across studies, which integrated the majority of reciprocal and refutational translations. We extracted suffering from social isolation from all studies, while the unique role of friends was only explicit in one study [49]. However, this construct resonated with extracts from other studies presenting how relationships with parents may challenge the process of getting on with life as a young person. Such challenges may appear when young people feel unable to live up to their parents’ expectations [42], or when roles are reversed and the child is positioned as a young carer of the parents [50]. Furthermore, a third extract showed that young people compared care staff with mothers during admissions, which implied care as well as the risk of being treated as a child [46]. Indirectly, these extracts indicated that socializing with friends involved less pressure [49] and enabled the sharing of youth activities and the chance to position oneself as an emerging adult in a life-disruptive situation. In order to improve transparency and illustrate the synthesis process, Table 3 presents examples of extracts translated into key metaphorical constructs while Table 4 illustrates reciprocal and refutational translations between studies.

### 2.6. Translating Studies into One Another

The first and last author made separate lists of the metaphors extracted from each study in order to chronologically compare and discuss the findings. The first author translated metaphors from the index study [40] to metaphors of the second study [41], thus developing initial reciprocal and refutational translations across these studies. Translations were compared with metaphors extracted from the third study [42], and so forth. This process continued consistently until all papers included in the study were compared, and a meta-interpretive synthesis was produced. During translations, the context of each study was carefully considered to interpret the findings in the context of varying disciplinary affiliation, cultural settings, sampling method, or other contextual specificities. A comparison of metaphors across studies involved considerable adjustment, merging and renaming in order to define key constructs. Subsequent to translations across studies, the first author reread all studies to extract paragraphs elaborating or explaining key metaphors. Starting with the index study, the first author wrote memos for each study to develop a common storyline. Alternative readings were discussed with the co-authors.

## 3. Synthesis of Included Studies

A common storyline across the included studies is that young people suffer from disconnectedness from activities and relationships with their age peers while experiencing life-disruptive mental distress, and that many of them make efforts to reconnect in various ways. In the phase of synthesizing translations, theoretical perspectives from other fields of research can be applied to refine interpretations [26,51]. The majority of translations in this meta-ethnography concern young people’s relationships with others, which inspired us to apply young people’s ‘belonging work’ as a lens to explore young adults’ recovery and identity processes during times of life-disruptive mental distress. The belonging work perspective derives from the field of sociology of youth and presents young people’s work to belong or disconnect while negotiating and shaping identity [23]. Researchers are guided by questions as to what actors do to belong, which actions are acceptable or expected, and who will be allowed to belong [52]. Hence, exploring belonging also involves exploring unbelonging [53]. Young people’s belonging work is explored within the dimensions of time, place and people [23,52,54]. As the included studies involve little interpretation of the meaning of places, this synthesis focuses on young people’s belonging work in the dimensions of time and people.

### 3.1. Expectations of Progression in Youth Contrast with Stagnation during Psychosis

Young people’s belonging work involves belonging within the dimension of time [54] including adaption to the movements or rhythms of their generation [52]. Within the dimension of time, it appears relevant to explore the existing social expectations for youth that young people navigate while performing belonging work.

A predominant reciprocal translation in this meta-ethnography is social expectations of progression in youth [49], according to which young people are expected to take certain actions to progress towards adulthood. The metaphors that translated into ‘expectations of progression in youth’ were “age-related milestones” [41], p. 260, “developmentally normative functioning” [45], p. 550 and “age-appropriate developmental tasks” [42], p. 430. Across all studies, expectations of increased independence stand out as a central metaphor for maturation. A key task is to emancipate from parents [40], which involves independent living [45] and the development of closer relationships with peers than parents [42]. Furthermore, the completion of education and establishment of a career [42] stand out as developmental tasks, as well as finding a partner and having children [44]. The authors of the index study suggest that young people with experiences of life-disruptive mental distress face the same social expectations as other young people [40]. No studies contradict this notion. Based on these translations, we argue that, while experiencing early psychosis or schizophrenia, young people recognize and navigate the same social expectations for young adulthood as other young people, as illustrated by MacDonald et al.:

“Consistent with their developmental stage, participants in this study also identified that issues related to separation-individuation, decreasing dependency on parents, and working towards personal autonomy were important” [40], p. 140.

Emancipation from parents, however, was absent in the Chinese study, even though a prompting interview question directly suggested “living away from parents” as an indicator of recovery [44], p. 3. In this perspective, expectations of young people’s independent living may particularly characterize a Western context. 

In summary, young people’s belonging with time in a Western cultural context involves progression towards independence and monitoring whether goals have been achieved and tasks have been completed in relation to education, career and the establishment of social relationships with people of a similar age. These social expectations encompass a framework for a refined understanding of why young people suffer from disconnectedness in times of life-disruptive mental distress and how they attempt to reconnect while striving for recovery. In opposition to the expectations of progression in youth, psychosis is associated with stagnation (Table 4). The majority of studies interpret psychosis as interruptive and a challenge to normal youth development (Table 4). Young people’s experiences tend to reflect this notion, as Huckle et al. illustrate:

“(…) participants described an experience of stagnation during their illness, as symptoms forced a period of absence from ‘normal life’ while peers continued to progress with expected milestones. Participants seemed to indicate a sense of being left behind with less things in common with their friends than before their illness (…) The period of illness seemed to ‘pause’ progress through life stages, including education and training opportunities, romantic relationships, and having children” [49], p. 9.

This extract indicates that young people are aware of the social expectations of progression in youth, yet feel unable to keep up during times of life-disruptive mental distress. A study of teenagers by Cogan et al. suggests that comparisons of oneself with the progression of age peers start early [48]: teenagers with experiences of psychosis grieve their own disabilities even before they are expected to live up to expectations regarding independent living, education and employment. Considering the social expectations of normal youth progression, they interpret their current situation as a threat to their future. However, three studies suggest that young people are often able to remain optimistic regarding recovery due to their young age, naivety and short-term experience of life-disruptive mental distress (Table 4). However, as the majority of the synthesized studies exclude young people who are severely impacted by mental distress or with problems related to use of drugs and alcohol, it is unclear whether this further challenged group share the experience of youth optimism.

### 3.2. Feeling Isolated, Lost and Left Behind

While the previous section showed how young people struggle with living up to the social expectations of youth to belong with their time and generation, this section presents how young people suffer from unbelonging with people during times of life-disruptive mental distress. Across the included studies, young people struggle with social isolation, and this stands out as a key challenge. We explore why social isolation occurs during early psychosis and the dynamics that reinforce disconnection and challenge belonging.

Young people’s initial experiences of psychosis often involve immense changes in their perceptions of themselves and the world. They feel that something is wrong, the world appears overwhelming, threatening, and unsafe, and they may find that they lose contact with reality, trust in themselves, and connectedness with others (Table 4), as illustrated by the following extract:

“The participants identified two factors that contributed to their social isolation: the loss of the friends they had before the onset of psychosis and difficulty in engaging in new friendships. According to the participants, the second factor—difficulty in engaging in new friendships—related to social and cognitive skills” [42], p. 428.

Sometimes, isolation is imposed on young people; when psychotic experiences force them to pause education or employment, their contact with other young people in such environments becomes limited [42]. Sometimes, young people are rejected by their friends or age peer network, who may perceive their changed behavior as strange or unusual [49].

The utmost form of imposed social isolation occurs when young people are admitted to mental hospitals, as connections with the outside world are further limited by the restriction of movements in and out of this environment:

“(…) participants experienced the ward as like being ‘in prison’, ‘on lock-down’, and ‘cut off from the world’. This conveys a sense of confinement, restriction and segregation. Viewing hospital as a prison evokes images of being trapped and having limited agency (...) It is reasonable to speculate that some aspects of the restrictive nature of inpatient environments might be experienced as particularly negative for young adults precisely because they are at a stage of life where they are only just beginning to achieve independence from institutional spaces and structures (e.g., schools) in their day-to-day lives” [46], pp. 237–238.

Contrary to the purpose of mental hospitals, Fenton et al. consider that admission may increase isolation and lead to traumatic experiences for young people [46].

They may also choose to isolate themselves while experiencing psychosis to pass as “normal” and avoid rejection and stigmatization. Stigma is present in the majority of studies; however, this theme is presented in great detail in the Chinese study [44], in which participants concealed their diagnoses of psychosis or schizophrenia from everyone but their close family. Some acquired ‘fake’ diagnoses to avoid stigma, and some found that biological explanations of their mental distress reduced their self-blame. However, no participants in this study experienced overt discrimination. Participants’ preoccupation with stigma and their extensive endeavors to adjust to normality may be related to Chinese culture; however, as this study was the only one based on a focus group, the expanded focus on stigma may also express what young people prefer to discuss with each other in focus groups compared to individual interview conversations with researchers.

The study by Hirschfeld et al. suggests that coming forward and connecting with others during times of crisis is a particular challenge to young men striving to adjust to expectations of masculine behavior and bottling up their problems in order to be seen as tough:

“He felt nobody understood, and that all he could do was bottle up problems at this time. For him the sense of being alone and misunderstood was bottled up and became a significant factor in the development of his psychosis. Possibly the cultural notion that men, especially young men need not talk about feelings or weaknesses was felt strongly by this young man (…). This could well be both a gender and age-related struggle that is magnified by the young man’s fears about how he will be received in light of his psychosis” [41], p. 259 and 261.

Hirschfeld et al. refer to a discursive polarization of gender roles associating masculinity with rationality and femininity with emotionality and madness, which may explain the reinforcement of these young men’s struggles with emotional openness and connectedness with others [41].

Furthermore, young people’s social withdrawal during experiences of psychosis may reflect a tendency to silence oneself as an act of care towards one’s parents:

“Nearly all participants stated that talking about their psychosis experiences was personally and relationally risky, and while some tried to initiate conversations, others avoided talking to family about their mental health altogether. For some, this included keeping their mental health status a secret or minimizing it significantly; for others, it was avoiding revisiting ‘bad’ experiences, even when they felt it might help them. Only one participant described a supportive discussion about psychosis with a family member. Most felt their mental health experiences were burdensome, and so they silenced themselves, perhaps at least partially, as an act of care for others” [50], p. 657.

This tendency appears characteristic among young people experiencing insufficient parental support, as only three out of ten participants in this study reported parental support during crises, while the remaining participants were either positioned as young carers of parents or felt pushed into independence by their parents.

As an ultimate act of withdrawal, some young people consider and attempt suicide when experiencing psychosis [41,43]. Suicidal thoughts and plans may be more widespread than suggested by translations of this meta-ethnography, as several studies excluded participants severely affected by mental distress or using drugs or alcohol [40,42,44,46,48]. Young people’s suicidal ideations and suicide attempts underline the importance of understanding young people’s recovery in order to intervene and prevent self-harm. While this section has explored young people’s experiences of unbelonging during psychosis, the following section presents young people’s work to belong with people; their attempts to maintain or establish connections particularly to age peers on their path to recovery.

### 3.3. Young Adults’ Recovery Involves Belonging with Other Young People

To overcome social isolation, young people’s recovery in psychosis involves social reconnection, particularly with their age peers. This section presents translations of how these young people understand recovery, and the dynamics involved in young adults’ recovery processes.

Although young people combine personal, social, psychological, clinical and existential notions of recovery in subjective understandings of the phenomenon, we translate a common desire across studies to get on with life like any other young person, as highlighted by MacDonald et al.:

“At this point in their recovery, the participants expressed that they were now thinking about getting on with their lives like any other young person, specifically negotiating separation-individuation issues with parents (...). The essence of the experience of these young people’s social relationships was their struggle to integrate their psychotic experiences with their experience of being young adults” [40], p. 138 and 141.

Young people’s wish to get on with life corresponds with social expectations of progression in youth, while it is opposed to metaphors related to psychosis such as being left behind, in stagnation or stuck (Table 4). The struggle of recovery implies that experiences of stagnation are difficult to integrate for a young person trying to keep up with expectations of youth progression [40]. 

In order to move on, young people make efforts to engage in youth activities and achieve youth goals, as illustrated by Huckle et al.:

“Five participants highlighted a desire to ‘get life back on track’, in terms of management of symptoms, housing, employment and/or education. For some participants this was perceived as integral to having the confidence to meet new friends and progress socially (…). When describing recovery, participants described a role for friends that was unique and could not be filled by family members. In particular, participants highlighted pressure from family members and unsolicited advice regarding recovery (…). In terms of aiding recovery, participants made sense of their friends’ involvement as providing support and distraction. It seems that doing everyday activities (such as playing football) helped participants to re-build an identity outside of the illness” [49], p. 12 and 15.

Huckle et al. highlight how young people strive for belonging with age peers through attempts to synchronize themselves with the activities and life rhythms of other young people to “get life back on track”. Hence, friendships play a unique role in young people’s recovery, and parents are not able to substitute friends [49]. Engagement in youth activities with friends tends to distract young people from rumination [49] and promote a meaningful integration of mental distress into their life story and identity [40]. Furthermore, being with friends involves less pressure than spending time with parents [49]. While parental support is crucial to young people during times of life-disruptive mental distress [40], Fenton et al. present an explanation of why belonging primarily with adults does not support young adults’ recovery process of moving on like any other young person. This is illustrated in a quote interpreting young people’s social contact with psychiatric staff during admission:

“(…) the service-users noticed that a set of parent-child scripts were activated. In some respects, such scripts might be helpful (e.g., as here, invoking nurture and care), and in others we could speculate that they may not (e.g., there is the potential for the ‘parent’ role to undermine independence, or overlook individuality). Simon described feeling ‘looked after’, and missed the staff once he left (he went back to visit the staff and say thank you). Although Mark also felt looked after, he did not feel actively cared for by staff and felt his individuality was overlooked” [46], p. 237.

Fenton et al. emphasize a set of parent–child scripts, assigning roles to young people and adults as, respectively, recipients and providers of care. This perspective appears relevant, as many young people still live with their parents and are still ‘being parented’ during times of life-disruptive mental distress [46]. Adopting the perspective of shaping identity through belonging, parents and professional caregivers reflect young people’s positions as children, while friends of similar age reflect young people’s positions as emerging adults. In summary, young people experiencing psychosis strive to belong with age peers in order to recover and move on with life like any other young person. This may explain why psychiatric staff’s support of young people’s recovery is absent in most of the included studies and scarcely discussed in all but one study [43], even though young people acknowledge the importance of the professional support [40,43,47].

Another adult represented in the included studies is the adult co-inpatient, and young people’s confrontations with adult inpatients in distress involve further complexities. While relationships between service users often enable mutual support, confrontations with adult inpatients’ distress can jeopardize young people’s recovery optimism, as argued by Fenton et al.:

“Whilst we must acknowledge that psychiatric hospitalization can be a threat to anyone’s identity, we must also emphasise the centrality of this concern in the accounts of these young people. Emerging adulthood is a time of crucial psychosocial development, individuation, memory formation, and identity construction (Harrop and Trower, 2003). Both psychosis itself and intense exposure to its potential consequences (via hospitalisation in an environment peopled by older and more experienced adults in distress), present substantive threats to this process, as we can see in the accounts above” [46], p. 238.

The experience of adult inpatients in distress confronts young people with the risk that they may themselves face long-term experiences of psychosis. According to Fenton et al., this poses a threat to young people’s identity construction and hopes for the future [46].

### 3.4. Forming Identity Positions of Growth and Disability during Psychosis

Consistent with the ways in which the included studies present young people’s reflections on identity, we perceive identity as constructed through social interactions (Table 4). In order to explore young people’s social identity processes and their chance of shaping a positive sense of self, this section presents how the identity positions of growth and disability are made available to young people during psychosis, and how these positions may enable or challenge belonging work.

Nine of the included studies explore the experiences of young people several years following the onset of psychosis, which provides further insight into the recovery trajectories over time (Table 2). While some participants report continuous experiences of incompetence, social isolation and distress [41,42,48], others experience an integration of mental distress with their identity and feel able to reconnect socially [41,43,44]. In retrospect, some even regard psychosis as life-enhancing [44]. Two identity positions stand out across the included studies, corresponding with the metaphors of progression and stagnation: personal growth and continuous experiences of incompetence and disability. Personal growth appears to occur in young people with the experience that psychosis has been integrated into their life stories and view of self in a positive way [41,44,45,47]. From this perspective, young people may believe that psychosis has provided them with valued life lessons, enhancing their awareness of their own values and ability to establish meaningful relationships and enjoy life.

According to refutational translations, young people may achieve personal growth through either maturation or social validation (Table 4). Personal growth through maturation is based on the idea of psychosis as a magnifier of ordinary coming-of-age struggles. The suggestion is that young people may grow out of psychosis as time passes. Hirschfeld et al. illustrate this ‘coming-of-age’ perspective as follows:

“…finding more meaning in existence through their individual experiences, and so their sense of handling the world around them and themselves matures. They come through the emotional challenges of their life stage, which appear magnified by their experience of psychosis and reflect on changes in their confidence and self-understanding. In this theme there are indications that the passage of time has allowed young men to reflect on the meaning their psychotic experiences has had in their lives, particularly in their relationships to themselves and others” [41], p. 260.

This idea that personal growth and recovery occur with the passage of time corresponds with the analytical reflections of Romano et al. regarding how recovery develops ‘over time’, noting that young people “used the recovery process as a ‘call to action’ when over time they developed new skills, made better choices and established more fulfilling relationships” [43], p. 248. The interpretation of personal growth through maturation implies an understanding of recovery as a personal process that develops with the progression of time.

The second perspective, personal growth achieved through social validation, is presented in a psychology dissertation [47]. A key interpretation of this dissertation is that young people’s identity processes are highly sensitive to experiences of validation or invalidation from others. Social validation of how young participants make sense of their experiences of life-disruptive mental distress appeared to improve their access to their personal potential, to promote their reconnection with others, and to reinforce their empathy and skills such as artistic talents [47]. From this perspective, developing a positive sense of self while experiencing psychosis in young adulthood is primarily a social process.

When encountering invalidation, young people tend to develop invalidating beliefs about themselves. This appears particularly common when others do not acknowledge the way young people make sense of psychosis:

“Participant Blue had a felt sense that viewing himself as disabled limited his capacity to reach toward his personal purpose and potential. Participants who did not access their personal potential had a felt sense that psychosis had delayed or permanently interrupted their personal growth process (…) Non-affirming relationships did not offer participants the time, space, and support they needed to develop validating beliefs about psychosis. Participants experienced this lack of support as offering them a narrow degree of freedom to make meaningful sense of psychosis. Participant K7 sensed this lack of support as encouragement to accept a belief that psychosis was a debilitating life experience (…) A restricted sense of identity incorporates psychosis as a negative influence on participants’ identity construction, giving them a sense they could not access their personal potential. Participant B12 experienced psychosis as turning him into a broken person” [47], p. 76, 77 and 81.

Hayden-Lewis’ interpretation of how some young people feel disabled or broken due to social invalidation corresponds with the key interpretations of Roy et al., who reported that the young participants experienced more handicap-creating situations than competence situations during psychosis (Table 4). It also corresponds with some young people’s understanding of recovery as a process of restoring a broken self, as presented by Windell et al. [45].

Given the prominent meaning of social interactions provided in many translations of this meta-ethnography (Table 4), recovery and the shaping of a positive sense of self may develop over the of passage time, yet these processes are deeply embedded in young people’s experiences of social validations and belonging.

### 3.5. Lines-of-Argument Synthesis: Navigating Relational Complexities in the Process of Recovery

In the final phase of this meta-ethnography, translations are brought together in order to create a new social explanation of the processes of young adults’ recovery and identity formation. Figure 2 integrates the reciprocal and refutational translations presented in Table 4 into a common storyline [36]. We employ a lens of belonging work focusing on young people’s navigation in webs of relationships [52] to illustrate the complexities that young people face during the processes of recovery and identity formation when situated in early adulthood.

While integrating translations in Table 4, Figure 2 illustrates how young people struggle to reconnect with others and break out of social isolation while experiencing psychosis. However, their work to belong with people involves multiple complexities: with age peers, young people appreciate long-lasting friendships, yet they face and fear rejection and stigma from their friends. They enjoy the social closeness of sharing drugs, yet shun friendships based on such habits in order to avoid further mental distress. They strive for reciprocity and benefit from the mutual support they share with other peer service users; however, they shun the stigma attached to this group. They seek closeness in romantic relationships, but extensive intimacy with one person may entail distance from others. Altogether, these young people face paradoxical complexities when performing belonging work to reconnect with age peers.

Young people’s relationships with adults involve other complexities. Positioning oneself as an emerging adult involves emancipating oneself from adults in caring roles. However, young people need sufficient parental support to develop positive views of emancipation from their parents. Despite their need for parental support, some young people silence themselves as an act of care to avoid burdening their family [50]. Finally, although belonging with peer service users may imply mutual support, facing adult co-inpatients in distress during admissions may reinforce young people’s fear of the future.

In summary, young people struggle to navigate the complexities of belonging within the dimensions of time and people to achieve social inclusion and recognition as emerging adults, which involves the establishment of meaningful relationships with age peers while attempting to avoid stigma and maintain hope. The above findings form a complicated web of interactions requiring careful considerations of social behavior that some young people handle entirely on their own while experiencing life-disruptive mental distress [41,50]. According to Hirschfeld, psychosis magnifies teenage struggles [41]. The navigation of relational complexities may not be unfamiliar to many young people without experiences of psychosis; however, this becomes an extensive task while struggling with life-disrupting mental distress. As suggested by Hayden-Lewis [47], these young people are in need of social validation to maintain the belief that their experiences of psychosis are not incompatible with a position of an emerging adult with adult rights. Corresponding with the perspective that belonging work involves paradoxes and ambivalence [23,52], young people’s acts of connection and detachment in their attempt to belong and avoid devaluation require complex navigation. This may explain why it can be so difficult to break out of social isolation during times of life-disruptive mental distress in young adulthood.

## 4. Discussion

This meta-ethnography synthesizes young people’s recovery and identity processes, as captured in empirical qualitative studies of young people’s experiences of life-disruptive mental distress and interpreted via a theoretical lens of young people’s belonging work. The study contributes with a social explanation [25] of how young people struggle with expectations of progression in youth, while facing stagnation and a loss of connectedness to age peers in times of life-disruptive mental distress. The struggle involves young people’s need to navigate the complexities of belonging with others, particularly their desire and effort to belong with other young people to achieve recognition in the position of a young adult. We interpret that young adults perceive recovery as the process of getting on with life like any other young person. Interactions with and validation from friends are key to this process, as this enables young people to form a positive self-image which is a vital element of recovery. Experiences of invalidation and socializing primarily with caring adults may keep young people stuck in the role of a child. Being able to position oneself as an emerging adult with adult civil rights supports young people’s empowerment.

The following section discusses how ‘getting on with life like any other young person’ and the unique role of friends during this process correspond with research in the field of recovery as a personal and social process. Finally, we discuss the idea of a “normal” transition to adulthood, indicated by authors of the synthesized studies in this meta-ethnography, as this idea constitutes the basis of young people’s comparison, identity formation and self-blame in times of life-disruptive mental distress.

### 4.1. Connectedness and Empowerment Enables a Positive Sense of Self

Corresponding with recent studies of young people’s recovery [55,56], this meta-ethnography argues that connectedness and social validation is equally important to young people’s experiences of recovery as moving on with life. In alignment with a concept analysis of recovery in young adulthood [56], the meta-ethnography suggests that the processes of connectedness and empowerment [57] are key to young people’s recovery. During a time of self-discovery, transition and acquisition of emerging adult roles [15,58], feeling connected, socially validated and empowered enables the formation of a positive sense of self. This intertwinement of connectedness, empowerment and identity formation in young people’s recovery corresponds with interpretations presented in a review of the evidence regarding social factors and recovery from mental health difficulties [5]. However, this meta-ethnography provides a social explanation of why these three processes are salient to young people and why friends play a unique role in young people’s recovery.

### 4.2. Getting on with Life with Friends 

While the vital importance of supportive social relationships in recovery is already well-explored [3,4,5,59], this meta-ethnography adds nuance to the unique role of friends in young people’s recovery. This involves sharing youth activities and synchronizing life rhythms as important elements of the recovery journey in young adulthood. It corresponds with studies underlining that young people suffer from reductions in their social network during experiences of life-disruptive mental distress, that poor friendships or social isolation tend to worsen young people’s mental health [60], and that friendships are essential to young people’s development of mental well-being and recovery [58,61]. Friendships appear more important to young people’s recovery than romantic relationships with age peers. Experiences of validation from and belonging with friends may involve fewer complexities than belonging with romantic partners, as romantic relationships may provide closeness with the partner yet increase distance from others [41]. This is in alignment with a study of the meaning of romantic relationships to young people diagnosed with psychosis stating that these young people fear to lose themselves with or to become too dependent on a romantic partner. They further express lacking experience with as well as resources for romantic engagement [62]. This emphasis on the unique role of friends in young people’s recovery calls for a discussion regarding what constitutes supportive and validating friendships. The meta-ethnography presented young people’s dilemma regarding their enjoyment of the intimacy of drug sharing with friends yet desire to distance themselves from these friends to stay away from drugs to avoid further mental distress. This is in alignment with a study on friendships and substance use arguing that the size of young people’s networks plays a role in their perceived quality of life. Hence, interventions may engage in changing the composition of young people’s social networks without reducing the size of them [63]. When building new networks, some young people benefit from belonging with other young peer service users. However, the meta-ethnography argues that while these relationships involve great potential for reciprocal support for some, others might prefer to avoid these networks. This is in alignment with a review of evidence regarding social factors and recovery, highlighting that some people prefer to engage in non-psychiatric networks to shape their identity outside the world of mental health [5]. 

The meta-ethnography illustrates how young people form their sense of self through comparison with others. This calls for a discussion of whether the idea of ‘normal’ transition to adulthood, that we generated though the meta-ethnographic synthesis, is in alignment with current analyses of the transition to adulthood among young people outside of a psychiatric context.

### 4.3. Multiple Routes to Adulthood and Recovery

The meta-ethnography suggests that, in a Western context, ‘normal’ progression to adulthood is associated with increasing independence, responsibility, and competence alongside the achievement of specific developmental milestones. Only one study included in this meta-ethnography normalizes the development of young people who experience psychosis by suggesting that still living with parents is rather common among young people in general [40]. The remaining synthesized studies do not discuss the linearity or pace of ‘normal’ transition to adulthood or to what extent ‘normal’ young people might also struggle with experiences of stagnation or social isolation. According to traditional views, the transition to adulthood involves progression through portals of education and work [54]. From this perspective, experiences of life-disruptive mental distress pose a crucial challenge to young people’s transition to adulthood, as struggles with educational and vocational achievements are frequent in such youth life situations [64,65]. However, transition research further argues that societal changes regarding individualization and detraditionalization, including increased social acceleration and general uncertainty about the future, have led to less linear routes of transition to adulthood [66,67,68,69], while accentuating young people’s personal responsibility to shape their own lives during their transition. Within youth research, the ongoing diversification and destabilization of transitions to adulthood has led to individualized transitions [69] described via metaphors as yo-yo-transition and reversible transition [70], p. 121. In this perspective, leaving the childhood home is no longer an irreversible process [71].

Life-disruptive mental distress does constitute an additional adversity in young people’s lives, however they are not alone in their struggle to belong. The destabilization of transitions towards adulthood creates an asynchronicity among young people, demanding intensified belonging work. Furthermore, a work culture comprising several varying jobs and unpredictable working hours makes it difficult to synchronize lives with friends and relatives, which increases experiences of social isolation among young adults in general [69,71]. 

These current perspectives inspire us to question the “normalcy” of a linear transition to adulthood that the young participants, as well as the authors of the synthesized studies, appear to imagine. As there are many routes to recovery [16,57], there may also be many routes to adulthood. In a time of destabilization and detraditionalization, it makes sense that young people yearn to belong with age peers in times of life-disruptive mental distress, as their life situations further reinforce experiences of deviation and desynchronization. However, it is regrettable when young people in life-disruptive mental distress compare themselves with idealized ideas of ‘normal’ young people’s uncomplicated belonging and linear transition to adulthood. Taking inspiration from the sociology of youth research, we find it fruitful to understand the experiences of young users of mental health services in the context of being a young adult in the current Western world. 

The extensive complexity of young people’s belonging work during their journeys of mental health recovery and transition to adulthood (Figure 2) may inspire further research in the field of young people’s recovery.

### 4.4. Strengths, Limitation and Reflections on Research Positions

A strength of the meta-ethnographic method is the inductive and repeated comparison of metaphors across the included studies and a thoroughly grounded integration of reciprocal and refutational translations into refined social explanations. This approach honors an authenticity criterion regarding fairness, defined as whether researchers have taken competing constructions into account [72]. The synthesis also involves several limitations related to perspectives that the included studies did not address. Firstly, the importance of material conditions for recovery was not elaborated, although studies of the social nature of recovery suggest that paid employment and material gain may enable social recognition and emancipation from experiences of illness [3,5]. Secondly, the included studies involved few interpretations of what particular places mean to young people’s recovery. Through a lens of belonging work, places are important contexts of identity formation. Hence, the meaning that places have in young people’s recovery appears relevant to pursue in further research. Thirdly, the included studies do not address young people’s agency in terms of how they may rebel against identity positions when diagnosed with a mental illness [73]. It appears relevant to pursue these perspectives in further research. Finally, a different theoretical lens or the inclusion of other empirical studies might have entailed different social explanations. The first author has a nursing background, but no clinical experience of psychiatric nursing. To honor the criterion of ontological authenticity [72], and to inform and sophisticate individual constructions, the first author obtained feedback from an interdisciplinary research group and a reference group of people with personal experiences of psychosis and recovery in young adulthood.

## 5. Conclusions

According to this meta-ethnography, social isolation is a key reason for young people’s misery while experiencing early psychosis or schizophrenia. It is a central element in young people’s recovery processes of moving on with life like other young people that they experience reciprocal friendships and share activities to synchronize their life rhythms with age peers. Belonging with validating friends can support young people’s processes of making sense of their experience of psychosis and guide them during transition to adulthood. This suggests a potential for young people to build positive self-images and empowerment together to avoid feeling lost and left behind while comparing oneself with notions of “normal” transition to adulthood.

### Implications for Practice

Parents and professionals can promote the recovery of young people experiencing life-disruptive mental distress by offering stable and continuous care, while recognizing and supporting young people’s need to belong with other young people. When adjusted to young people’s individual preferences and needs, this involves protecting existing friendships and connections to youth environments as well as supporting the potential to make new friends. A recently published scoping review examines supportive interventions for young people’s maintenance or building of friendships during times of mental distress and the risk of social isolation [61]. The review suggests that there are positive short-term benefits to friendship-building interventions while stressing the need for more insight into the meaning of friendships to young people with experiences of life-disrupting mental distress.

## Figures and Tables

**Figure 1 ijerph-20-06653-f001:**
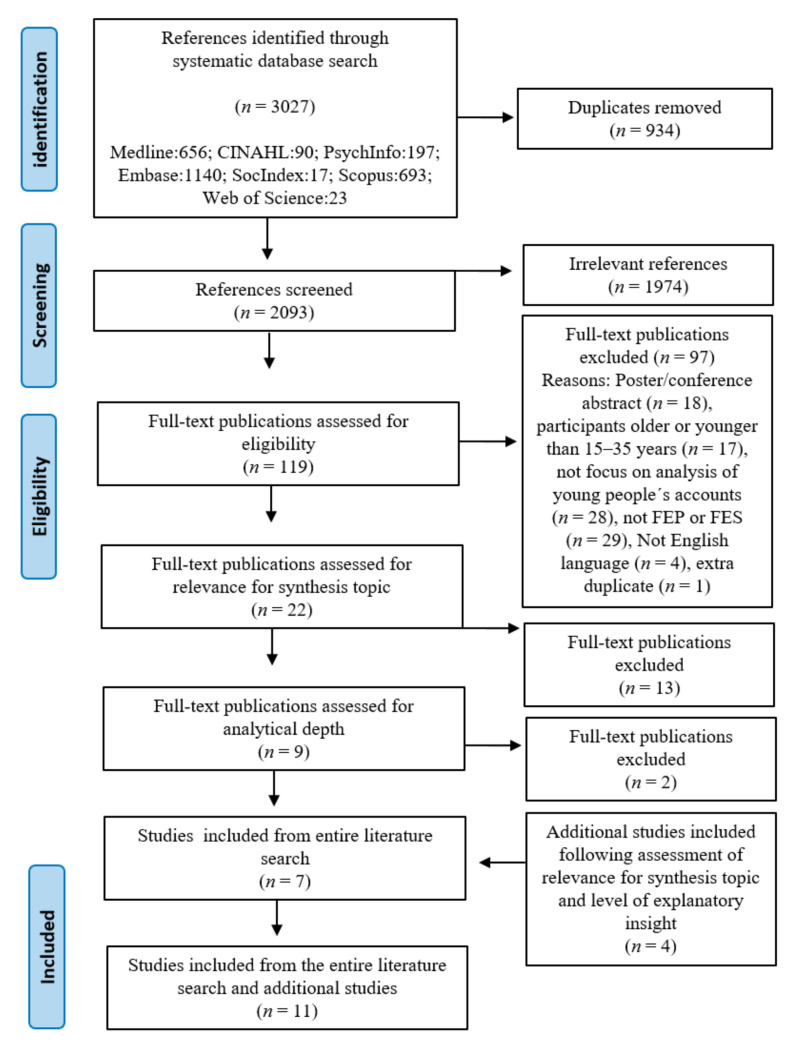
Flow chart illustrating the literature search process. FEP = first episode psychosis, FES = first episode schizophrenia.

**Figure 2 ijerph-20-06653-f002:**
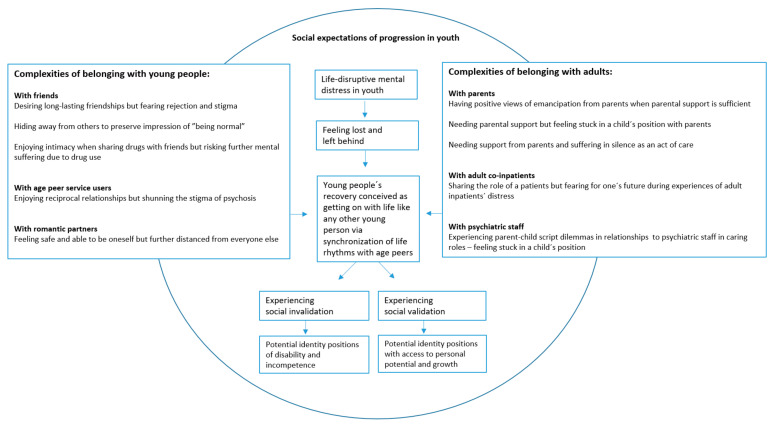
Lines-of-argument synthesis. Complexities of belonging with people to achieve a suitable social position during recovery from psychosis in young adulthood.

**Table 1 ijerph-20-06653-t001:** Criteria of inclusion and exclusion.

Inclusion Criteria	Exclusion Criteria
English or Nordic languages. Qualitative empirical studies. Interpretations of young people’s experiences. Early experiences of psychosis or schizophrenia. Vast majority of participants aged 15–30 years. Interpretations of youth, psychosis/schizophrenia and recovery including identity formation.	Published before the year 2000. Conference papers, posters, reviews and expert opinions. Primarily or entirely quantitative studies. Primarily organizational studies (‘pathways to care’). Interpretations of non-youth perspectives. Studies of young people ‘at risk’ (not yet diagnosed). Minority of participants with early psychosis/schizophrenia. No finding publications and topical surveys.

**Table 2 ijerph-20-06653-t002:** Characteristics of the studies included in the present meta-ethnography.

Year and Authors	Field of Research	Study Context	Sample	Recruitment Details	Data Collection Method	Methodological Approach	Aim	Findings
2005 Mac-Donald et al. [40]	Psychology	Australia Recovery Group Program in an early psychosis treatment	6 participants 19–25 years 5 males, 1 female	Experiences with social relationships following psychosis onset. Interviewed 1–2 years following first admission.	Multiple semi-structured in-depth interviews	Phenomenology	To explore young people’s experiences of social relationships following first episode of psychosis in order to facilitate their social relations.	*“Participants in this phenomenological study were juggling a desire to be involved in normal adolescent activities with wanting to be with people who are accepting and understanding. They described not keeping in touch with their old friends associated with usual changes in friendships across the life cycle, along with a real or perceived concern about potential rejection, and a wish to leave behind harmful lifestyles and activities. The essence of the experience of these young people’s social relationships was their struggle to integrate their psychotic experiences with their experience of being young adults.”* (p. 144)
2005 Hirschfeld et al. [41]	Psychology	UK Users of psychiatry recruited through keyworkers	6 participants 19–29 years All male	Interviewed 3–5 years following first episode of psychosis.	Semi-structured in-depth interviews	Constructivist Grounded Theory	To explore the subjective experiences of young men during psychosis and the meaning the experiences have for them.	*“The analysis has produced four themes common to all the accounts: experience of psychosis, immediate expression of psychotic experiences, personal and interpersonal changes, and personal explanations.”* (p. 262)
2009 Roy et al. [42]	Occupational Therapy	Canada The youth psychosis clinic of the Hôpital Sacré-Coeur in Montreal, Quebec	19 participants 18–30 years 16 male, 3 female	Interviewed on arrival at youth psychosis clinic prior to inclusion in rehabilitation component of the program. Interviewed within 5 years following first episode of psychosis.	Semi-structured interviews	Undefined	To explore the competence and handicap-creating situations experienced and perceived by young adults living with recently onset schizophrenia in their daily roles and activities.	*“The participants experienced more handicap-creating situations than competence situations. The themes included a diminished quality of relationship with parents, social isolation and difficulties in work and academic performance, as well as poor access to education. The perception of the participants on each of these themes is elaborated.”* (p. 424)
2010 Romano et al. [43]	Nursing	Canada First episode psychosis program	10 participants 19–30 years 5 male, 5 female	Interviewed twice 1–3 years following initial treatment for first episode of psychosis.	In-depth semi-structured interviews	Constructivist Grounded Theory	To explore how individuals describe their process of recovery following first episode of schizophrenia and how identified individuals, e.g., family members, describe their perceptions of and roles in the participant’s recovery process.	*“The results provide a substantive theory of the process of recovery from FES that is comprised of the following phases: ‘Who they were prior to the illness’, ‘Lives interrupted: Encountering the illness’, ‘Engaging in services and supports’, ‘Re-engaging in life’, ‘Envisioning the future’; and the core category, ‘Re-shaping an enduring sense of self’, that occurred throughout all phases. A prominent feature of this model is that participants’ enduring sense of self were reshaped rather than reconstructed throughout their recovery.”* (p. 243)
2011 Lam et al. [44]	Psychiatry	China/ Hong Kong The Early Assessment Service for Young People with Psychosis (EASY) in Hong Kong	6 participants 23–29 years 3 male, 3 female	Interviewed from 1.4 to 6 years following first episode of psychosis.	Focus group	Non-specified qualitative analytical method	To explore subjective recovery experiences following first-episode psychosis and the meaning attached to experiences of illness and treatment.	*“(…) four major themes: the meaning of psychosis and psychotic experience; the meaning of recovery; stigma; and having an optimistic view of recovery (…) Participants’ view of recovery was broader than that often held by psychiatrists.”* (p. 1)
2012 Windell et al. [45]	Psychology	Canada Follow-up assessment at a specialized early interventionservice	30 participants 25.9 years (mean) 23 male, 7 female	Interviewed 3–5 years following initiation of treatment for psychosis.	Semi-structured interviews	Phenomenology	To explore personal definitions of recovery among individuals recently treated in a specialized early-intervention service.	*“A majority of individuals considered themselves to be recovered. Responses indicated that recovery is a multidimensional experience and is often a personalized and achievable goal at this early stage in treatment. Individuals described recovery as improvement in one or more of three domains: illness recovery, psychological and personal recovery, and social and functional recovery. There was variation in the extent to which individuals perceived that recovery involved alleviation of symptoms and elimination of underlying vulnerability to illness.”* (p. 548)
2014 Fenton et al. [46]	Psychology	UK Early Intervention Services	6 participants Mean 24.5 years 5 male, 1 female	Interviewed 1–1.5 years following first episode of psychosis.	Semi-structured interviews	Phenomenology	To explore experiences of hospitalization during first episode of psychosis.	*“Findings describe fear and confusion at admission, conflicting experiences of the inpatient unit as both safe and containing, and unsafe and chaotic, and the difficult process of maintaining identity in light of the admission.”* (p. 234)
2015 Hayden-Lewis [47]	Psychology	USA Early Assessment and Support Alliance identifying young people with psychosis	7 participants 19–24 years 6 male, 1 female (1 person changed sexual orientation during the research project)	No information on length of participants’ experiences with psychosis.	Intensive semi-structured interviews	Constructivist Grounded Theory	To explore how young adults who experience psychosis and schizophrenia develop their sense of identity.	*“Data analysis generated the central category of identity construction, called “making sense of psychosis”. Analysis also illuminated the properties of making sense of psychosis, which were called developing beliefs about psychosis and degrees of freedom. These properties and their dimensions influenced and were influenced by the contexts exploring relationships and relating to personal potential. The general consequence of participants’ process and experience of making sense of psychosis was called “becoming who I really am”, which best described participants incorporating psychosis into their sense of identity. The potential for a reciprocal action process existing between making sense of psychosis and becoming who I really am was also explored.”* (Preface)
2019 Cogan et al. [48]	Psychology	UK Early Intervention Service in the Scottish National Health Service	10 participants 16–18 years 5 male, 5 female	Interviewed following 1–5 years of contact with the mental health system.	Semi-structured interviews	Thematic Analysis	To explore adolescents’ personal understandings and experiences of recovery during early onset of psychosis and how subjective experiences of living with psychosis have impacted their self-identification.	*“Qualitative analysis of adolescents’ accounts revealed how recovery from psychosis involves working with individual explanatory frameworks concerning uncertain identities and status ambiguity, a decrease in reference points and unfavorable social comparisons (emphasizing loss, grief and self-criticism).”* (p. 169)
2021 Huckle et al. [49]	Psychology	UK Early intervention service for psychosis in London	14 participants Mean: 25 years7 male, 7 female	Purposive sampling regardless of engagement with services or level of social functioning. Interviewed 0.5–2.8 years after first referral to psychiatric services.	Semi-structured interviews	Thematic Analysis	To explore experiences of friendship of young people during first episode of psychosis, focusing especially on any perceived changes in their friendships or approach to peer relationships as a result of psychosis.	*“Identified themes included the loss of social contacts because young people developing psychosis withdrew and because friends withdrew as illness developed. Regarding recovery, a unique role was identified for friends, and participants were often making conscious efforts to rebuild social networks. Mental health services were viewed as having a limited direct role in this.”* (p. 1)
2021 Boden-Stuart et al. [50]	Psychology	UK Early psychosis services	10 participants 18–23 years 5 male, 5 female	Purposive sampling. Interviewed following attendance at early-intervention service for 1–3 years.	Relational mapping interview	Multimodal hermeneutic phenomenological approach	To explore how young people experiencing early psychosis ‘map’ and describe their experiences and understandings of their family relationships, and how they have related to their psychosis and recovery.	*“Findings explore the participants’ accounts of how they love, protect, and care for their families; how they wrestle with family ties as they mature; and their feelings about talking about their mental health with loved ones, which was typically very difficult.”* (p. 646)

**Table 3 ijerph-20-06653-t003:** Examples of how central extracts inspired the construction of metaphors.

Metaphor	Examples of Extracts that Inspired Metaphor Construction
Expectations of progression towards independence in youth	*“They described their personal experiences of adjusting to age-related milestones like going onto further and higher education, finding work, increasing their sense of independence and personal responsibility, developing sexual relationships and changing relationships with friends and parents.”* [41], p. 206 *“(…) the developmental tasks facing young adults: continuing studies; finding a life partner; having children.”* [44], p. 4 *“Participants reported tensions between moving away and staying connected with family, wrestling to balance their needs for closeness and separateness and the expectations of others with regard to maturity and recovery.”* [50], p. 656
Psychosis as stagnation and challenge to ‘normal’ youth development	“*As expected the onset of a psychotic illness has a major impact on a young person’s life and heightens the challenges of fulfilling the developmental roles, and of engaging in activities and relationships with other people.”* [40], p. 139 *“The majority experienced a reduced capacity to participate in activities (e.g., school, work), engage with friends, and became increasingly dependent on family.”* [43], p. 247 *“Six participants described an experience of stagnation during their illness, as symptoms forced a period of absence from ‘normal life’ while peers continued to progress with expected milestones. Participants seemed to indicate a sense of being left behind with less things in common with their friends than before their illness.”* [49], p. 9
Suffering from social isolation following psychosis	*“Participants gave first-hand descriptions of feelings of loneliness, isolation and demoralization prior to joining the* [group] *program.”* [40], p. 130 *“Losses were experienced by the adolescents in multiple domains of their lives (peer and romantic relationships or educational achievements) following their episode of psychosis.”* [48], p. 174 *“The reported findings demonstrated experiences of loss of social contacts as a result of first episode psychosis, resulting from either participants or friends withdrawing, and highlighted the intense effort and vulnerability involved in building new relationships for this client group.”* [49], p. 13
Young people’s recovery as a social/interactive and active process	*“Participants identified the importance of having the support of others as they re-engaged in life particularly highlighting the fact that their recovery did not occur in isolation.”* [43], p. 248 *“Perception of “being able to do something about it” included identifying potential (personalized) avenues for agency and control of the experience and the experience of being able to enact these strategies. This component of recovery often involved specific lifestyle changes to support one’s recovery.”* [45], p. 550 *“Across all three dimensions (friends, parents, fellow service-users), participants described interpersonal connections which made a positive difference for their insight and recovery.”* [46], p. 239
Young people’s recovery as getting on with life like any other young person	*“At this point in their recovery, the participants expressed that they were now thinking about getting on with their lives like any other young person, specifically negotiating separation-individuation issues with parents.”* [40], p. 138 *“Five participants highlighted a desire to ‘get life back on track’, in terms of management of symptoms, housing, employment and/or education.”* [49], p. 12 *“Some individuals indicated that social recovery was (or would be) experienced as establishing independent adult living, emphasizing that being recovered involved competence and maturity as a young adult.”* [45], p. 550
The unique role of friends in young people’s recovery	*“Twelve participants (40%) specifically identified social participation such as peer relationships and romantic attachments in their recovery definitions. Although relationships with family were often described as playing a crucial role in recovery, these relationships were only rarely described as a specific component of the meaning of being recovered.”* [45], p. 550 *“There was a shared idea that friends could provide support for participants to get back involved with everyday life, and to resume activities which might have been difficult to initiate alone (…) Participants experienced friendships differently to family relationships in the recovery process, which seemed to be related to a sense of less pressure or expectation.”* [49], p. 10 *“(…) the data illustrated the strengthening of existing relationships and participants identified a unique role for friends in the process of recovery.”* [49], p. 13

**Table 4 ijerph-20-06653-t004:** Reciprocal and refutational translations of key metaphors.

Reciprocal Translations: Youth	Psychosis	Recovery	Identity
Expectations of progression in youth [40,41,42,43,45,46,49,50]	Changed relationships with oneself and the world during experiences of psychosis [40,41,43,45,48]	Young people’s recovery means getting on with life like any other young person [40,42,43,49]	The social formation of identity [42,45,47,49,50]
Youth as a process of increasing independence [40,41,43,45,46,49] presupposing competence and maturity [46]	Psychosis/schizophrenia as stagnation and a challenge to ‘normal’ development in youth [40,41,42,43,44,46,47,48,49]	Recovery as an active and interactive process [40,41,42,43,44,45,46,47,48,49,50]	Making personal sense of psychosis and integrating experiences with identity [40,43,45,47,48]
Youth as an identity process [47,48]	Social isolation [40,41,42,43,44,47,48,49], due to loss of friends [40,41,42,47,48,49], loss of capacity to participate [43,47,48], stigmatization [40,41,42,43,44,46,47,48], self-stigma [41,44,47,48,49] and silencing oneself as an act of care [50]	Recovery as the ability to function among others [44,45,47]	
Hopes and expectations of increasing independence [40,41,43,45,46,49]	Experiencing difficult emotions [41,43,44,46] including thoughts about dying [41,43]	Need for reciprocity: The importance of giving and receiving support [43,46,47,49,50]	
Optimism in youth [43,44,45]		Being with friends is more important to move on with life as a young person than being with parents or staff [45,49]	
**Refutational Translations:** **Youth**	**Psychosis**	**Recovery**	**Identity**
	Psychosis as illness [40,42,43,44,48,49] vs. psychosis as experience [41] vs. psychosis as experience and illness [46,47,50]	Personal growth occurs through maturation vs. through social validation [41,43] vs. [47]	Personal growth as an identity position following recovery vs. continuous feelings of incompetence, disability and restricted identity [41,44,45,47] vs. [42,47,49]
		Peer service users provide support and potential friendships yet belonging with this group is associated with stigma [40,48,49] vs. [46,48,49]	Engaging with friends enables reflection of the role as emerging adult, yet it involves the risk of rejection and role loss [40,44]
		Engaging with friends enables participation in youth activities, yet it involves the risk of rejection and loss of participation [40,44]	The need to appear ‘normal’ to protect one’s identity from stigma may imply isolating oneself to avoid disclosure of psychosis causing further distress [40,44]
		A positive view of emancipation from parents presupposes sufficient parental support, while complicated relationships with parents involve ambiguous feelings about emancipation [40,42,50] vs. [41,42,50]	A broken vs. reshaped identity following psychosis [45,47] vs. [43,47]
		Extensive intimacy with a partner is soothing yet it involves further distance from others, which may evoke further social isolation [42]	
		Relationships with peer service users may enable mutual support, yet adult inpatients’ distress during admissions threatens young people’s identity construction and hopes for the future [40,42,45,46,48] vs. [46]	
		Young people may feel cared for yet stuck in the child’s position in relationships with staff who assume parental roles [40,43] vs. [46]	

## Data Availability

All relevant data are included in the paper and Appendix A.

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
