# Peer review of "Social Processes of Young Adults’ Recovery and Identity Formation during Life-Disruptive Mental Distress—A Meta-Ethnography"

_ijerph, 2023, doi:10.3390/ijerph20176653_

Round 1

Reviewer 1 Report (New Reviewer)

The manuscript reports a qualitative synthesis of the literature on recovery and identity formation among young adults with early psychosis. I appreciate the authors’ effort in this detailed work, including literature search, analysis and synthesis, and interpretation of the quotes and themes. The whole study was well-executed, following established guidelines and protocols. The findings reflect the challenges faced by young adults with early psychosis and supplement current quantitative findings on recovery from early psychosis. I have some comments for the authors’ consideration.

 Introduction:

-        The Introduction may benefit from a summary of what the literature says on the correlates/ factors of recovery in individuals with early psychosis. A wealth of findings from quantitative studies have identified these correlates/ factors, which would be informative to the topic and set the stage for the current qualitative study.

-        The Introduction could be further developed with the challenges specific to early psychosis (vs other mental disorders) in the process of recovery. These may include psychotic symptoms like paranoia and social anhedonia, as well as internalized stigma and discrimination. This point should be highlighted throughout the manuscript.

-        The authors seemed to use ‘youth’ and ‘young adults (& emerging adulthood)’ inter-changeably throughout the manuscript. Please be precious about the developmental period referred to in the text, as there is non-overlap between these two stages. This may define the scope of the review, as well as the included studies and subsequent findings and conclusion (e.g. some developmental tasks/ events may be more relevant to adulthood, rather than late adolescence, like career).

-        The meaning of the phrase ‘new social explanations’ (line 101, p.3) is not clear. Please elaborate or revise.

Methods:

-        Please update the search, as the review is only limited to studies published by March 2021.

 Results:

-        I wonder if there how the management of psychotic symptoms and relapse would be important to the process of recovery. The findings current synthesis did not touch on these relevant issues to early psychosis, which I think are important to self-efficacy and empowerment.

-        For Figure 2, I am curious about the differentiation between ‘complexities of belonging with young people’ and that with ‘adults’, as there may be social figures in both categories (e.g. friends who are more mature in age, above 18). This may also be relevant to my third comment for the Introduction above.

 Discussion:

-        Further implications for practice from the findings could be discussed. For example, how do the findings inform current practices of case management? What are the roles of various clinicians and mental health professionals in the recovery from early psychosis? What are the proposed solutions to target social isolation/ exclusion to promote recovery? These are potential directions arising from the findings.

Author Response

Introduction

The Introduction may benefit from a summary of what the literature says on the correlates/ factors of recovery in individuals with early psychosis. A wealth of findings from quantitative studies have identified these correlates/ factors, which would be informative to the topic and set the stage for the current qualitative study.

Thank you for this comment. It has inspired us to rewrite the introduction section. We became aware that we had not managed to clarify the perspective and purpose of this study in the introduction section. We hope that we managed to clarify that this study explores social processes. We explore young people with psychosis or schizophrenia due to the life-disrupting function of these conditions. We acknowledge the importance of quantitative research involving correlations and psychiatrically defined symptoms. However, we do not consider this to be within the scope of this current study.

The Introduction could be further developed with the challenges specific to early psychosis (vs other mental disorders) in the process of recovery. These may include psychotic symptoms like paranoia and social anhedonia, as well as internalized stigma and discrimination. This point should be highlighted throughout the manuscript.

Thank you for this comment. Psychotic symptoms are important and of great meaning to the suffering human being. However, this study does not focus on clinical symptoms, but on recovery and identity as personal and social processes associated with psychosis/schizophrenia that we grasp as life-altering events. Therefore, we have not analyzed the meaning of specific symptoms, yet we do analyze stigma.

The authors seemed to use ‘youth’ and ‘young adults (& emerging adulthood)’ inter-changeably throughout the manuscript. Please be precious about the developmental period referred to in the text, as there is non-overlap between these two stages. This may define the scope of the review, as well as the included studies and subsequent findings and conclusion (e.g. some developmental tasks/ events may be more relevant to adulthood, rather than late adolescence, like career).

Thank you for this comment. It clearly improves this study. Throughout the study, we have now made an effort to clarify that this synthesis interprets young adults´ recovery even though one of the included studies involves a few minors. However, we have chosen to hold on to “young” as an adjective describing teenagers as well as young adults. We make this choice with reference to Arnett´s conceptualization of the transition from childhood to adulthood as socially constructed referring to certain markers of maturity (education, family building, etc.)  Arnett suggests that today this transitional period continues well into young people´s twenties due to Western cultural changes. Furthermore, experiences of life disruptive mental suffering appear to extend this period of transition. We present Arnett´s approach in the introductory section. 

The meaning of the phrase ‘new social explanations’ (line 101, p.3) is not clear. Please elaborate or revise.

Thank you for this comment. According to the founders of the meta-ethnographic method, Noblit & Hare, the purpose of the method is to seek new social explanations through grounded and comparative interpretations (Noblit & Hare, 1988, p .23). We hope that a thorough revision of the introduction section has clarified that this study explores social and cultural understandings of young people´s recovery. The phrase "seeking social explanations" can be considered synonymous with this.

Methods

Please update the search, as the review is only limited to studies published by March 2021.

Thank you for this comment. A meta-ethnography can be based on a systematic literature search, but this is not a requirement, according to the founders of the method (Noblit & Hare, 1988). We conducted a systematic literature search in order to select the most interesting studies for this synthesis. However, a meta-ethnography is not a review. The founders of the method suggest that meta-ethnographies are repeated at intervals of decades due to dynamic cultural and social changes. Therefore, we have not updated the search. On the other hand, we have tried to meet the expectation of being updated by introducing other recent studies of young people's recovery in the discussion section.

Results

I wonder if there how the management of psychotic symptoms and relapse would be important to the process of recovery. The findings current synthesis did not touch on these relevant issues to early psychosis, which I think are important to self-efficacy and empowerment.

Thank you for this comment. Psychotic symptoms and coping strategies are important. The study synthesizes interpretations of selected studies and the young people in these studies did not elaborate much on how they managed symptoms and relapse. One common reaction was to either integrate the experiences of psychosis into one´s life story and identity or to seal over the experiences. This is presented in the meta-ethnography p. 18 l. 1-6 and p. 19 l. 30-33.

For Figure 2, I am curious about the differentiation between ‘complexities of belonging with young people’ and that with ‘adults’, as there may be social figures in both categories (e.g. friends who are more mature in age, above 18). This may also be relevant to my third comment for the Introduction above.

Thank you for this comment. It is very interesting how young people might relate to older friends. Unfortunately, none of the young people who shared their stories in the selected studies described friendships with people who were older than them.

Discussion

Further implications for practice from the findings could be discussed. For example, how do the findings inform current practices of case management? What are the roles of various clinicians and mental health professionals in the recovery from early psychosis? What are the proposed solutions to target social isolation/ exclusion to promote recovery? These are potential directions arising from the findings.

Thank you for this comment. In the section Implications for Practice we have made an effort to express that young people would benefit from the support of professionals when it comes to maintaining and building lasting social relationships with age peers and to inspire clinical practice we refer to a recently published scoping review examining supportive interventions for young people´s friendships. We hope that this could be useful for various clinicians.

Reviewer 2 Report (New Reviewer)

I am interested in the topic and clinically work in a related area.
I thought the introduction was good and I was looking forward to reading through the paper.
That said, the meta-ethnography was interesting to read.
However, does this paper represent science?
A few suggestions:

*Mandatory*
-On line 132 you casually mention "X" without any introduction. What is "X"? I am not familiar with papers written like this. 
-Also on that same line you casually mention "building block search" -- you need to introduce it and tell me exactly what you mean by this.
-X continues throughout the paper

*Optional*
-On line 109 you casually mention that you will be following an interpretivist paradigm -- does this mean that science can be disregarded?
Instead of writing long verbose conclusions (which are interesting to read but again don't really fully fall into being science) you really should try to analyze the data a little bit more quantitatively -- I am not asking for a lot, just a tiny bit of showing some quantitative relationships.  I think the paper can be published even if you don't do this, so it is not mandatory, but you will have a better paper if you try to do a tiny bit of quantitative work (really... you will.... you might also realize better whether your verbal conclusions are justified or not).

Author Response

Comments and Suggestions for Authors:

I am interested in the topic and clinically work in a related area. I thought the introduction was good and I was looking forward to reading through the paper.

That said, the meta-ethnography was interesting to read. However, does this paper represent science?

Thank you for this comment. The meta-ethnographic method is a well-established, well-developed, and frequently used methodology within health science to explore social processes through meta-interpretation.

Bondas T, Hall EOC. Challenges in approaching metasynthesis research. Qual. Health Res. 2007, 17(1), 113–121.

France EF, Uny I, Ring N, Turley RL, Maxwell M, Duncan EAS, Jepson RG, Roberts RJ, Noyes JA. Methodological systematic review of meta-ethnography conduct to articulate the complex analytical phases. BMC Med. Res. Methodol. 2019, 19(1), 35.

France EF, Cunningham M, Ring N, Uny I, Duncan EAS, Jepson RG, m.fl. Improving reporting of meta-ethnography: the eMERGe reporting guidance. BMC Med. Res. Methodol. 2019, 19(1):25.

Mandatory

On line 132 you casually mention "X" without any introduction. What is "X"? I am not familiar with papers written like this.

Thank you for this comment. X was meant to anonymize the authors´ contributions to the study. Now every X has been replaced by either the first, second, or last author.

Also on that same line you casually mention "building block search" -- you need to introduce it and tell me exactly what you mean by this.

Thank you for this comment.

The correct expression is a search involving a “building block strategy”, thank you for making us aware of this. In order to explain what this strategy implies, we have rearranged the section so that the explanation of search involving a building block strategy follows directly after this search methodology is mentioned:

“Assisted by an information specialist [35], the first author conducted a search involving a building block strategy [36] in 7 databases between January and March 2021 (Figure 1). The search combined five search blocks. Four blocks, ‘youth’, ‘psychosis’, ‘recovery’ and identity’, were defined with reference to the aim and research question of the study, while the fifth block limited the search to qualitative studies. We identified search terms in thesauruses of each database and free-text words through examination of keywords of relevant publications. Free-text words were applied across databases. Given the study’s specific interest in youth, we exclusively searched for youth and free-text synonyms of this term in the title field. As young people’s experiences of recovery are an eligible subject of research within a range of disciplinary fields, the search was conducted in several databases, including Medline, CINAHL, PsychInfo, Embase, SocIndex, Scopus, and Web of Science. Due to broad perceptions of recovery and identity as both personal and social processes, various synonyms were used (Table of search string from Embase in the supplementary file)”

X continues throughout the paper

Thank you for this comment. All the Xs have been replaced with the first, second, or last author.

Optional

On line 109 you casually mention that you will be following an interpretivist paradigm - does this mean that science can be disregarded?

Thank you for this comment. We are uncertain if we understand this comment correctly. We understand the comment as if science is referred to as science within the positivist paradigm. Our answer relies on our understanding of the comment.

Research within the interpretivist paradigm is preoccupied with cultural and social dynamics i.e. in human relationships. They seldomly follow the strictness of cause and effect,   but can be understood as situated and possibly changing over time. We do not mean to disregard research conducted within the positivist paradigm. However, research within the interpretivist paradigm is able to generate a different kind of insight into social and cultural processes.

Instead of writing long verbose conclusions (which are interesting to read but again don't really fully fall into being science) you really should try to analyze the data a little bit more quantitatively -- I am not asking for a lot, just a tiny bit of showing some quantitative relationships.  I think the paper can be published even if you don't do this, so it is not mandatory, but you will have a better paper if you try to do a tiny bit of quantitative work (really... you will.... you might also realize better whether your verbal conclusions are justified or not).

Thank you for this comment. As the meta-ethnographic method is a qualitative approach within the interpretivist paradigm, we have not analyzed the material by use of quantitative methods.

We consulted quantitative research colleagues who recommended that we rewrote the introduction section to clarify the purpose of the study and reduce the readers´ impression of verbosity. They further recommended that we deleted the following sentence, which we have done:

P.10 l. 16-19:

“This involves potential limitations. With reference to Cogan et al. the excluded group “…may be more inclined to experience difficulties in terms of developing a stable sense of self. Preliminary discussions with senior clinical staff who assisted with recruitment reported this to be the case” [55, p. 175].

We hope that this revision has altogether improved the study.

Round 2

Reviewer 2 Report (New Reviewer)

I accept the authors' responses.
Thank you for you the very important changes to the Introduction section. Given this new Introduction I am willing to accept the remainder of the paper which follows.
Again, I don't think the full potential scientific value of the topic has been covered by this paper, but it is acceptable for publication.

This manuscript is a resubmission of an earlier submission. The following is a list of the peer review reports and author responses from that submission.

Round 1

Reviewer 1 Report

Review Report

 The research objects are relatively focused; the methodological steps are complete and standardized; the correlation and classification of key concepts and themes, and the synthesis of translation basically achieve higher-order refinement. The article also puts forward valuable service practice and policy recommendations.

Some problems exist in the structural level, layout and content of the paper, which could be checked and improved.

I. About the structure

1. The format of the headings is inconsistent. Sections from “1. Introduction” “2. Method” to “7. Synthesizing Translations”   have serial numbers, and then there are no serial numbers for the sections of Discussion and Conclusion. Please check.

2. The structural hierarchy of some sections is not clear; sections2, 3, 3.1, 4, 5, and 6 are not at the same level as section1 and section 7. It seems better to integrate section2, 3, 3.1, 4, 5, and 6 as components of “Method” to sort out.

3. Based on the context, “Table 2” should be revised to “Table 1” and adjusted to the section “5 determining how study are related"”. The original Table 1 should be adjusted after “7. Synthesis Translation”.

4. L687-698 also seems to discuss the limitations of the study, while some of the content in the “Limitation” section is more like responding to the limitations. If “Limitation” and implications for practice can be placed into the Discussion section, a secondary title might be set in the first half of the “Discussion”.

II. About the content

1. Research questions require corresponding literature review. Previous literature on relevant research topics could be reviewed to highlight the necessity and important value of this study.

2. In the “Method” section, it might be added to explain why this study is suitable for metaethnographic analysis, or explain the advantages of applying metaethnographic analysis to the study of this topic.

3. Section “3. Deciding what is Relevant”: It is not uncommon to have only one subtitle (“3.1 quality appraisal”). In order to clearly present the implementation steps of the study methodology, the content of this section can be adjusted, e.g. 3.1 Selecting and locating relevant studies, 3.2 Inclusion and exclusion criteria, 3.3 Quality appraisal.

5. Section “5. Determining how Studies are Related” only presents the relevance of each study in terms of disciplinary affiliation, object of study, and research scenario, but the relevance of the most important key concepts and themes is not described and illustrated.

6. The process of translation into one another is not fully explained in the section of “6 Translating Studies into One Another”. After L194-195 “X compared these translations with metaphors extracted from the third study and so forth”, a similar statement as follows could be added: this process consistently continued until all papers included in the study were combined and a synthesis of the lead author's interpretation was produced.

7. It would be clearer if the structure of the section “7. synthesis translation” is presented in categories of “reciprocal translation”, “refutational translation”, and “lines of argument”. In addition the content of this section seems to need to be explained in the study aims or questions of this paper.

8. The text in the “Discussion” section needs to be distinguished from the text in the sections of “Abstract” and “Conclusion” so as to avoid mutual repetition.

III. Others

1. The age ranges of the study subjects in L64 and L103 are not consistent with those in figure 1.

2. L125-L128,“This is consistent with recommendations of a maximum of 10 (Sandelowski et al., 1997) to 12 studies for synthesis (Paterson et al., 2001) as the wealth of information and the context of each publication may be lost with a larger number of studies (Bondas & Hall, 2007). ”  It has better to confirm whether the number of papers included in the study “10-12 is “maximum or “minimum, or “ a smaller number. The number of papers included in the study can be between 40 and 100 according to the previous literature on the application of meta-ethnographic methods (Sattar, R., Lawton, R., Panagioti, M. et al. Meta-ethnography in healthcare research: a guide to using a meta-ethnographic approach for literature synthesis. BMC Health Serv Res 21, 50 (2021). https://doi.org/10.1186/s12913-020-06049-w).

3. L510, The year of the cited literature seems to be incorrect, it should be 2015 according to Table 1, please check.

4. L580-581, the discussion in this sentence “Altogether, young people face a considerable ambivalence when performing belonging work to reconnect with age peers during psychosis” would be better if it could indicate the literature materials.

Author Response

I. About the structure

1

The format of the headings is inconsistent. Sections from “1. Introduction” “2. Method” to “7. Synthesizing Translations”   have serial numbers, and then there are no serial numbers for the sections of Discussion and Conclusion. Please check.

Thank you for your comment. I will address them point by point. All sections are now equipped with serial numbers.

2

The structural hierarchy of some sections is not clear; sections 2, 3, 3.1, 4, 5, and 6 are not at the same level as section1 and section 7. It seems better to integrate section 2, 3, 3.1, 4, 5, and 6 as components of “Method” to sort out.

Thank you for your comment. We have changed the structural hierarchy according to the suggestions.

3

Based on the context, “Table 2” should be revised to “Table 1” and adjusted to the section “5 determining how study are related"”. The original Table 1 should be adjusted after “7. Synthesis Translation”.

Thank you for this comment, we have revised the tables and adjusted them to the suggested sections.

4

L687-698 also seems to discuss the limitations of the study, while some of the content in the “Limitation” section is more like responding to the limitations. If “Limitation” and implications for practice can be placed into the Discussion section, a secondary title might be set in the first half of the “Discussion”

Thank you for this comment.

We have gathered content and rearranged the sections of discussion, limitations, conclusion and implications for practice in an order that we hope is more appropriate.  

II. About the content

1

 Research questions require corresponding literature review. Previous literature on relevant research topics could be reviewed to highlight the necessity and important value of this study.

Thank you for this comment.

Research specifically exploring the phenomenon of youth recovery is still sparse. Yet, we have found and inserted additional studies of recovery in youth and include their findings in the discussion section.

2

In the “Method” section, it might be added to explain why this study is suitable for metaethnographic analysis, or explain the advantages of applying metaethnographic analysis to the study of this topic.

Thank you for this comment. We have added following sentence: “This study positions itself within the interpretivist paradigm and hence we have chosen an interpretive type of meta-analysis. In contrast to narrative or systematic reviews aggregative in nature, the purpose of interpretive literature syntheses is to develop meta-analytical qualitative explanations innovative in nature. The meta-ethnographic synthesis is a well-developed method including inductive comparative processes of translation. It is eligible when qualitative empirical studies have thoroughly explored a specific field of interest” (Britten 2002). The approach appears relevant, as we managed to find no other interpretive literature syntheses on youth recovery during psychosis.

3

Section“3.  Deciding what is Relevant”: It is not uncommon to have only one subtitle (“3.1 quality appraisal”). In order to clearly present the implementation steps of the study methodology, the content of this section can be adjusted, e.g. 3.1 Selecting and locating relevant studies, 3.2 Inclusion and exclusion criteria, 3.3 Quality appraisal.

Thank you for this comment. We have applied the suggested sub-headlines.

4

Missing

5

Section “5. Determining how Studies are Related” only presents the relevance of each study in terms of disciplinary affiliation, object of study, and research scenario, but the relevance of the most important key concepts and themes is not described and illustrated.

Thank you for this comment. We have presented the most important translations in a table showing text extracts that inspired metaphor construction in order to reinforce transparency of the translational processes.

6

The process of translation into one another is not fully explained in the section of “Translating Studies into One Another”. After L194-195 “X compared these translations with metaphors extracted from the third study and so forth”, a similar statement as follows could be added: this process consistently continued until all papers included in the study were combined and a synthesis of the lead author's interpretation was produced.

Thank you so much for this suggestion. We have included the statement in the section.

7

 It would be clearer if the structure of the section “7. synthesis translation” is presented in categories of “reciprocal translation”, “refutational translation”, and “lines of argument”. In addition the content of this section seems to need to be explained in the study aims or questions of this paper.

Thank you for this comment. We have arranged reciprocal and refutational translations in table 3 to provide an overview of all conducted translations.

8

The text in the “Discussion” section needs to be distinguished from the text in the sections of “Abstract” and “Conclusion” so as to avoid mutual repetition.

Thank you for this comment. A new abstract has been developed.

III. Others

1

The age ranges of the study subjects in L64 and L103 are not consistent with those in figure 1.

Thank you for this comment.

Very few individuals of the altogether 124 were younger than 18 years or older than 30 years (only one study included some adolescents and one other study included a person aged 34. Hence we have now written “a vast majority of participants were aged 18-30 years. 

2

L125-L128,“This is consistent with recommendations of a maximum of 10 (Sandelowski et al., 1997) to 12 studies for synthesis (Paterson et al., 2001) as the wealth of information and the context of each publication may be lost with a larger number of studies (Bondas & Hall, 2007). ”  It has better to confirm whether the number of papers included in the study “10-12”  is “maximum”  or “minimum”, or “ a smaller number”. The number of papers included in the study can be between 40 and 100 according to the previous literature on the application of meta-ethnographic methods (Sattar, R., Lawton, R., Panagioti, M. et al. Meta-ethnography in healthcare research: a guide to using a meta-ethnographic approach for literature synthesis. BMC Health Serv Res 21, 50 (2021). https://doi.org/10.1186/s12913-020-06049-w).

Thank you for this comment.

We chose the eleven studies for synthesis because they honored the criteria of relevance for topic and cut-off level of explanatory insight. We have added this explanation in the section of criteria of inclusion and exclusion.

3

L510, The year of the cited literature seems to be incorrect, it should be 2015 according to Table 1, please check.

Thank you for this comment, we have corrected the year to 2015 in the table.

4

L580-581, the discussion in this sentence “Altogether, young people face a considerable ambivalence when performing belonging work to reconnect with age peers during psychosis” would be better if it could indicate the literature materials.

Thank you for this comment. The sentence sums up figure 2, which we constructed by integrating translations of table 2 in a common storyline. We have added an explanation of this in the line of argument section and we hope that this will clarify how figure 2 builds on translations of extracts from included studies.

We hope that we have addressed all your useful comments and feedback and we look forward to your response.

Reviewer 2 Report

The authors applied a meta-ethnographic approach to investigate personal recovery and identity formation in a very specific population: young people suffering from early psychosis. Since the experience of psychosis has a strong impact on both psychological and sociological points of view, especially for the age group in question, this topic makes a field of notable interest. Some minor revisions are necessary and reported as follows.

Abstract 

The abstract needs rewriting:

-it is written that “the synthesis presents four constructs” but there are five of them and it would be better to avoid the numbers (1,2,3..);

-an english revision is needed;

-hypothesis and result should be reported more concisely and clearly.

Introduction

The introduction is well-written and effective in providing the theoretical framework of the study. However, since the research regards early psychosis and the authors properly refer to schizophrenia, it would be suggested to underline how the disease itself affects the patient’s identity and social life and to include a brief specific reference to the already-mentioned early psychosis interventions. This would help to underline the importance of both authors’ aims and findings. A minor english revision is required.

Methods

3. Deciding what Is relevant

Figure 1 should be revised as follows: “extra duplicate=1” should be replaced with “extra duplicate (n=1)”.

Discussion

The work is wide and comprehensive, all the sections are well organized and provide a clear and detailed explanation of the process followed by the authors step by step. Anyway, this section might risk appearing unfocused: it would be suggested to report the three key constructs of this synthesis all together at the beginning of the section, and then they should be discussed one by one as already done, to provide a more punctual discussion. 

Since the concepts highlighted are of great interest in clinical settings, a brief synthetic insertion concerning the implications of the author’s findings in clinical practice could enrich and complete the discussion: considering your findings on youth recovery and identity formation which new elements the modern clinical settings should provide? Conversely, which could hinder recovery?

Moreover, lines 687-698 could be moved to the “limitations” and a more precise definition of what are, in your opinion, the future perspective of research in this field should be provided.

Conclusion

The conclusion is well-written and organized.

This is a very stimulating and broad work, congratulations to all authors.

Author Response

Abstract: The abstract needs rewriting

1

It is written that “the synthesis presents four constructs” but there are five of them and it would be better to avoid the numbers (1,2,3..)

We have rewritten the abstract according to suggestions.

2

An English revision is needed;

Thank you for this comment. We have arranged English revision.

3

Hypothesis and result should be reported more concisely and clearly.

Thank you for this comment, we have adjusted the research question and rewritten the conclusion.

Introduction

1

The introduction is well-written and effective in providing the theoretical framework of the study. However, since the research regards early psychosis and the authors properly refer to schizophrenia, it would be suggested to underline how the disease itself affects the patient’s identity and social life and to include a brief specific reference to the already-mentioned early psychosis interventions. This would help to underline the importance of both authors’ aims and findings. A minor english revision is required.

Thank you for this comment. We have added that research suggests that psychosis and schizophrenia is associated with a loss of self or a distorted sense of self which accentuates exploring youth recovery and identity formation during psychosis

Methods

1

3. Deciding what Is relevant

Figure 1 should be revised as follows: “extra duplicate=1” should be replaced with “extra duplicate (n=1)”.

Thank you for this comment. We have corrected the figure according to the suggestion.

Discussion

1

The work is wide and comprehensive, all the sections are well organized and provide a clear and detailed explanation of the process followed by the authors step by step. Anyway, this section might risk appearing unfocused: it would be suggested to report the three key constructs of this synthesis all together at the beginning of the section, and then they should be discussed one by one as already done, to provide a more punctual discussion. 

Since the concepts highlighted are of great interest in clinical settings, a brief synthetic insertion concerning the implications of the author’s findings in clinical practice could enrich and complete the discussion: considering your findings on youth recovery and identity formation which new elements the modern clinical settings should provide? Conversely, which could hinder recovery?

Moreover, lines 687-698 could be moved to the “limitations” and a more precise definition of what are, in your opinion, the future perspective of research in this field should be provided.

Thank you for this comment. We have carefully rewritten the discussion based on three key constructs: youth recovery as getting on with life like any other young person, the unique role of friends to youth recovery and building identity through comparison with notions of “normal” development /transition towards adulthood in youth. We hope that this new discussion and order of the final sections of the article is sufficiently clear and improves the reading experience. 

Conclusion

1

The conclusion is well-written and organized.

This is a very stimulating and broad work, congratulations to all authors.

Thank you for all your useful comments, they have improved the study.

We hope that we have addressed all your useful comments and feedback and we look forward to your response.

Reviewer 3 Report

This study attempts to clarify the social process of youth recovery and identity formation during early psychosis. As a theme, it is very interesting. However, it is a very arbitrary method of validation, in which qualitative research is integrated qualitatively to create a story. While this is interesting in terms of forming a research concept, it is questionable as to whether or not it is a valid study. 

You employ a method called meta-ethnography, but how does this differ from qualitative integration through so-called systematic reviews? 

In addition, please provide the names of the seven databases in the text as well as in the figures.

Please indicate inclusion/exclusion criteria in the text, not just in the supplement file.

Also, you have prepared the manuscript ignoring the instructions of the journal. It is a basic matter. For example, the way you describe References is different from the journal instructions. Please refer to the instructions and correct it. Also, necessary items such as ethical guidance, conflict of interest, funding, and author contributions are missing.

L191-203 I don't know what X means. 

L99 appendix is a Supplement file?

Author Response

1

This study attempts to clarify the social process of youth recovery and identity formation during early psychosis. As a theme, it is very interesting. However, it is a very arbitrary method of validation, in which qualitative research is integrated qualitatively to create a story. While this is interesting in terms of forming a research concept, it is questionable as to whether or not it is a valid study. 

Thank you for this comment.

As Noblit and Hare, the founders of meta-ethnography, suggest the metaethnography, alongside qualitative research in general, is understood as partial and positional, depending on reviewer´s perspective (Noblit & Hare 1988).

To strive for rigor we conducted an exhaustive and systematic search and selevted studies that included interpretations of youth, early psychosis/schizophrenia, recovery and identity. To enable the reader´s assessment of the study´s trustworthiness, we have strived for methodological and analytical transparency and we hope that, following this revision of the study, our development of reciprocal and refutational translations will stand out more clearly.  

2

You employ a method called meta-ethnography, but how does this differ from qualitative integration through so-called systematic reviews? 

Thank you for this comment. In the method section we have added following sequence in order to address this:

“This study positions itself within the interpretivist paradigm and hence we have chosen an interpretive type of meta-analysis. In contrast to narrative or systematic reviews aggregative in nature, the purpose of interpretive literature syntheses is to develop meta-analytical qualitative explanations innovative in nature. The meta-ethnographic synthesis is a well-developed method including inductive comparative processes of translation. It is eligible when qualitative empirical studies have thoroughly explored a specific field of interest (Britten et al 2002). The approach appears relevant, as we have managed to find no other interpretive literature syntheses on youth recovery during psychosis”.

3

In addition, please provide the names of the seven databases in the text as well as in the figures.

Thank you for this comment. We have added following sentence:

As young people´s experiences of recovery is an eligible subject of research within a range of disciplinary fields, the search was conducted a range of databases in Medline, CINAHL, PsychInfo, Embase,SocIndex, Scopus and Web of Science.

4

Please indicate inclusion/exclusion criteria in the text, not just in the supplement file.

Thank you for this comment. We have added a separate section with this subtitle and inserted a table of inclusion and exclusion criteria.

5

Also, you have prepared the manuscript ignoring the instructions of the journal. It is a basic matter. For example, the way you describe References is different from the journal instructions. Please refer to the instructions and correct it. Also, necessary items such as ethical guidance, conflict of interest, funding, and author contributions are missing.

Thank you for this comment.

We have adjusted the article in accordance with instructions of the journal.

We are not sure how to address ethical guidance, as we solely apply research literature in this study and have had no contact with interviewees.

Conflict of interest, funding, and author contributions was submitted alongside first draft, however the information was inserted elsewhere in IJERPH´s submission formular which may be why you did not receive them. We will add this information to the main document.

6

L191-203 I don't know what X means. 

Thank you for this comment.

X is inserted to anonymize authors of the study  during the review process. I would like to apologize that the anonymization was not consistent in the first version we submitted, as I did not replace all initials with X.

7

L99 appendix is a Supplement file?

Thank you for this comment.

Yes, this has been corrected.

We hope that we have addressed all your useful comments and feedback and we look forward to your response.